# Purification, Characterization and Evaluation of the Antitumoral Activity of a Phospholipase A2 from the Snake *Bothrops moojeni*

**DOI:** 10.3390/ph15060724

**Published:** 2022-06-07

**Authors:** Breno Emanuel Farias Frihling, Ana Paula de Araújo Boleti, Caio Fernando Ramalho de Oliveira, Simone Camargo Sanches, Pedro Henrique de Oliveira Cardoso, Newton Verbisck, Maria Lígia Rodrigues Macedo, Paula Helena Santa Rita, Cristiano Marcelo Espinola Carvalho, Ludovico Migliolo

**Affiliations:** 1S-Inova Biotech, Programa de Pós-Graduação em Biotecnologia, Universidade Católica Dom Bosco, Campo Grande 79117-900, MS, Brazil; brenoemanuelfarias@gmail.com (B.E.F.F.); apboleti@yahoo.com.br (A.P.d.A.B.); simonecsanches@gmail.com (S.C.S.); cardosoliveira20@gmail.com (P.H.d.O.C.); cristiano@ucdb.br (C.M.E.C.); 2Faculdade de Ciências Farmacêuticas, Alimentos e Nutrição (FACFAN), Universidade Federal de Mato Grosso do Sul (UFMS), Campo Grande 79603-011, MS, Brazil; oliveiracfr@gmail.com (C.F.R.d.O.); ligiamacedo18@gmail.com (M.L.R.M.); 3Embrapa Gado de Corte, Campo Grande 79106-550, MS, Brazil; newton.verbisck@embrapa.br; 4Biotério e Serpentário, Universidade Católica Dom Bosco, Campo Grande 79117-900, MS, Brazil; paulahsr@ucdb.br; 5Programa de Pós-Graduação em Bioquímica, Universidade Federal do Rio Grande do Norte, Natal 59078-970, RN, Brazil

**Keywords:** phospholipase A2, biochemical characterization, cytotoxic activity

## Abstract

Nature presents a wide range of biomolecules with pharmacological potential, including venomous animal proteins. Among the protein components from snake venoms, phospholipases (PLA_2_) are of great importance for the development of new anticancer compounds. Thus, we aimed to evaluate the PLA_2_ anticancer properties from *Bothrops moojeni* venom. The crude venom was purified through three chromatographic steps, monitored by enzymatic activity and SDS-PAGE (12%). The purified PLA_2_ denominated BmPLA2 had its molecular mass and N-terminal sequence identified by mass spectrometry and Edman degradation, respectively. BmPLA2 was assayed against human epithelial colorectal adenocarcinoma cells (Caco-2), human rhabdomyosarcoma cells (RD) and mucoepidermoid carcinoma of the lung (NCI-H292), using human fibroblast cells (MRC-5) and microglia cells (BV-2) as a cytotoxicity control. BmPLA2 presented 13,836 Da and a 24 amino acid-residue homologue with snake PLA_2_, which showed a 90% similarity with other *Bothrops moojeni* PLA_2_. BmPLA2 displayed an IC_50_ of 0.6 µM against Caco-2, and demonstrated a selectivity index of 1.85 (compared to MRC-5) and 6.33 (compared to BV-2), supporting its selectivity for cancer cells. In conclusion, we describe a new acidic phospholipase, which showed antitumor activity and is a potential candidate in the development of new biotechnological tools.

## 1. Introduction

Cancer is a generic term for several diseases characterized by abnormal cell growth, with the potential to invade and spread to other tissues, thus becoming one of the most important health problems and responsible for the third largest number of deaths worldwide. The high incidence of patients’ morbidity and mortality is triggered by several molecular and anatomical subtypes of cells, requiring specific treatment strategies [1].

In 2020, 19.3 million cases and 10 million cancer deaths were estimated worldwide, where the most diagnosed were: prostate (7.3% of new cases), colorectal (10%), lung (11.4%) and breast (11.7%) cancer. Lung and colorectal cancer are the deadliest types of cancer, accounting for 1.8 million and 800,000 deaths, respectively, representing nearly 28% of all cancer deaths [2].

In addition to the high mortality rates, anticancer therapies involve invasive procedures, such as catheter-administered chemotherapy, surgical removal procedures and the use of non-selective cytotoxic drugs. Thus, bioprospection for new active drugs for cancer therapy is one of the goals of biotechnology, which depends directly on the extraction and purification of toxins and secondary metabolites from microorganisms, plants and animals [3,4].

Snake venom toxin’s ability to cause toxicity is associated with its high number of different molecules acting on the cells and tissues. Despite their toxicological effects, these biological samples possess a complex mixture of different components, including peptides, proteins, enzymes, carbohydrates and minerals. However, isolated compounds, such as proteins and peptides, have the potential application as pharmaceutical agents. The enzyme groups frequently found are L-amino acid oxidases, serine proteases, metalloproteases and phospholipases A_2_ (PLA_2_) [5,6,7].

Due to their role in a large number of human inflammatory diseases, PLA_2_ presents a medical–scientific interest. The classification of this protein group occurs according to the site of hydrolysis, where PLA_2_ are enzymes that catalyze the hydrolysis of phospholipids at the sn-2 position, releasing free fatty acids, arachidonic acid and lysophospholipids. By consequence, they play important roles in the metabolism of dietary and structural lipids in cell membranes. Hydrolysis of the lipids in cell membranes leads to a loss of their structure, impairing their selective permeability. As a consequence of this activity, svPLA_2_ has a diverse antimicrobial activity, being able to act on cultures of pathogenic bacteria, fungi, protozoa, viruses and tumor lineages [8].

Previous reports demonstrated that *Bothrops* svPLA2 from snake venom presents antitumor activity. BthA-I-PLA2 from *B. jararacussu* showed antitumor activity in leukemia (Jurkat), human breast tumor (BR-3) and Ehrlich ascites (EAT) lines; BmooTX-I (*B. moojeni*) and MTX-I (*B. brazili*) indicating activity against leukemic lineage (Jurkat); and Myotoxin III (*B. asper*) demonstrating cytotoxicity in adrenal tumors. In addition, other molecules of svPLA2 presented antitumor activity, such as PLA2 RVV-7 (*Daboia russeli*) showing activity against melanoma (B16F10), MVL-PLA2 (*Macrovipera lebetina*) with activity against fibrosarcoma, melanoma, adenocarcinoma and leukemia, and F1 CTX (*Crotalus durissus terrificus*) indicating cytotoxicity against cervical and esophagus cancer cell lines [9,10,11,12,13].

In an attempt to seek new alternative therapies, nature is an inexhaustible reservoir of compounds with the potential to treat a wide spectrum of diseases. In this context, we report, for the first time, a purified phospholipase A_2_ isoform from *B. moojeni* venom (BmPLA2), which displayed anticancer activity against colorectal adenocarcinoma (Caco-2) and human rhabdomyosarcoma (RD) cell lines.

## 2. Results

### 2.1. Purification and Biochemical Characterization

*B. moojeni* crude venom was fractionated by molecular exclusion chromatography on a Sephacryl S-100 column, resulting in five peaks, named F1 to F5 (Figure 1A). Peak F3 was selected for the next chromatographic step, based on enzymatic assay and its protein profile on SDS-PAGE. When F3 was separated in a C18 column, four peaks were collected, named F3.1 to F3.4 (Figure 1B). The peak F3.4 was chosen according to the above-mentioned parameters. As the last purification step, F3.4 was resubmitted to the RP-HPLC C18 column (Figure 1C), confirming the presence of a single peak, named BmPLA2. The BmPLA2 was eluted with 52.4% of solvent B.

To determine the BmPLA2 intact mass, a sample was subjected to MALDI-ToF analysis, which showed a protein with 13,838.4 Da (Figure 1D). Furthermore, the combination of HPLC and MALDI-ToF confirmed the molecular homogeneity of BmPLA2, since a single peak was noticed in both analyses. Through Edman degradation, the first 24 amino acid residues from the BmPLA2 N-terminal sequence were sequenced. The obtained sequence was NH2-FKWQFEMLIMKIAKTSGFMFYSSY-COOH. This sequence was submitted to the NCBI Protein BLAST and three PLA2 sequences with high sequential similarity were found. The PLA2 BmooPLA2 (*B. moojeni*), Tgc-E6 (*Tantilla gracilis*) and D1E6b (*Cerrophidion godmani*), showed a 95, 77 and 77% similarity, respectively, (Table 1). The presence of 15 identical residues and five conserved substitutions suggests that BmPLA2 belongs to the PLA_2_ family. The most diverse region in the PLA_2_ was in the first three amino acids. However, for PLA_2_ from the *Bothrops* genus, Trp-3 occupied an identical position.

### 2.2. Enzymatic Activity

Enzymatic activity assays were carried out to monitor PLA_2_ activity in the fractions collected in the chromatographic steps (Figure 2). The substrate used, 4N3OBA, is specific for PLA_2_ activity. BmPLA2 showed the highest PLA_2_ activity when compared to the positive control and the fractions obtained in the chromatographic steps, presenting an activity that was 34% higher than the commercial PLA_2_.

### 2.3. Hemolytic Activity

The hemolytic assay indicated that BmPLA2 displayed a low activity against mouse erythrocytes, when compared to the Triton X-100, indicating less than 3% of hemolysis in a high concentration (37 µM).

### 2.4. Anticancer Activity

Cytotoxicity against MRC-5, BV-2, Caco-2, NCI and RD cells was determined by cellular viability assay using MTT. The CACO-2 and RD cells’ viability was significantly inhibited by BmPLA2, while the NCI lineage did not undergo a significant reduction in the evaluated concentrations. At the highest concentration used (9.25 µM), 70% of the inhibition was noticed for the Caco-2 cell viability (Figure 3). In addition, the RD strain showed an approximately 60% reduction in cell viability at all concentrations evaluated. In comparison, for MRC-5 cells, at 9.25 µM, BmPLA2 prompted a reduction in cellular viability of 70% and the BV-2 cells showed a reduction in cell viability of approximately 40% (at 9.25 µM). Against the NCI tumor lineage, it was observed that cell viability did not show any differences between the concentrations evaluated, reducing approximately 30% of the cell viability.

Based on the IC_50_ obtained for cancer and healthy cells, we calculated the selectivity index (SI) (Table 2). Using MRC-5 as a comparison, in the Caco-2 strains, the SI values were 1.85, while when comparing with the BV2, the SI values were 6.33, suggesting that BmPLA2 possesses selectivity for cancer cell membranes in comparison with healthy membranes. The SI values for the NCI and RD strains were not calculated because the IC_50_ of these strains could not be obtained. To obtain this value in the RD strain, it would be necessary to repeat the evaluation, using lower concentrations of the BmPLA.

## 3. Discussion

Snake venoms are a rich source of new potential therapeutic compounds, presenting a wide range of pharmacological activities. All venoms showing biotechnological potential should be investigated through protein purification methods. Thus, the isolation and functional characterization of components will provide a basis for understanding the venom mechanisms of action and/or future molecular design [14].

PLA_2_ has been successfully isolated from snake venom using different chromatographic techniques. A general consensus is held that the PLA_2_ purification strategies involve the combination of types of gel filtration reverse phase chromatography, such as for venoms from the *Bothrops* genus (BA SpII RP4 from *B. alternatus*, BaPLA2-I and BaPLA2- III from *B. atrox*), and PLA_2_ from another genus (Pgo K49 from *C. goodmani*, Cdcum6 from *Crotallus durissus cumanensis*, LmTX-I, LmTX-II from *Lachesis muta*). Beyond the classic strategy used here, other strategies combining gel filtration and ion-exchange, anion-exchange followed by cation-exchange, and ion-exchange followed by a reverse phase have been reported for PLA_2_ purification [15,16,17,18,19,20].

The acetonitrile percentage required for the elution of BmPLA2 was 52.2%, with a similar value for BaTX and BmTX-I, and a PLA_2_ from *B. alternatus* and *B. moojeni* was eluted with about 50% acetonitrile in RP-HPLC. The svPLA_2_ purification investigations, and svPLA_2_ using RP-HPLC with a C18 column, demonstrated that the percentage of acetonitrile needed to elute the molecule is close to 60%, for example, Bmaj-9 (*B. marajoensis*) and Cdr-13 (*C. durissus ruruima*). This reflects the similarity of the overall folding of the amino acids into the primary sequence, shared by snake PLA_2_ [21,22,23,24].

BmPLA2 showed a molecular mass of 13,838.4 Da, determined by mass spectrometry. The pattern of molecular mass of further PLA_2_ can be highlighted, such as PrTX-III, from *B. pirajai* (13,867 Da); Mtx-I, from *B. brazili* (13,870 Da); Myotoxin-IV, from *B. asper* (13,886 Da); BmooPLA2, from *B. moojeni* (13,601 Da); and a PLA isoform from *B. erythromelas* (13,656 Da). Furthermore, BmooPLA2 showed a 95% identity with BmPLA2 through an amino-terminal sequence (Figure 3). The amino-terminal sequencing of BmPLA2 showed homology with other snake PLA_2_, sharing the highest identity with BmooPLA2. These two types of PLA_2_ differed only by three amino acids: Phe^1^, Lys^2^ and Met^19^, which in BmPLA2 corresponded to Asn^1^, Leu^2^ and Leu^19^ [25,26,27,28].

Enzymatic activity with the specific substrate was performed to track the PLA_2_ through the purification process. Regarding PLA_2_ activity, the tracking might be performed by direct and indirect methods. First, the direct detection uses chromogenic substrates, such as 4N3OBA, which, when the hydrolysate releases a colorimetric product, is detected at 425 nm. As described in our results, BmPLA2 and the fractions assayed showed catalytic activity when compared to the positive control (bovine pancreas PLA_2_), which indicates that the specific substrate used was consumed by BmPLA2. Observing Figure 2, we noted that BmPLA2 presented catalytic activity superior to commercial PLA_2_, showing 34% greater activity, at the same concentrations.

The enzymatic activity of svPLA_2_ has been reported in previous works. Normally, PLA2 Asp^49^ has catalytic activity, as seen in sPLA2 from *C. durissus collilineatus* and PrTX-IIIB from *B. pirajai*. In contrast, the replacement of position 49 by a Lys^49^ indicates a reduction or even absence of catalytic activity of svPLA_2_, such as Basp-III (*B. asper*), BthTXI (*B. jararacussu*) or BnSP6 (*B. matogrossensis*). Thus, the enzymatic activity observed for BmPLA2 supports the hypothesis that position 49 contains an Asp residue (Asp^49^), due to its catalytic activity [29,30,31,32,33].

Low hemolytic activity observed for BmPLA2 (~3%) at the highest concentration evaluated (37 µM) is a common feature among svPLA_2_. BE-I-PLA2, a PLA_2_ from *B. erythromelas*, did not show hemolysis at the highest concentration tested (37.5 µM) and presented a greater catalytic activity than the evaluated control (bovine PLA_2_), at the concentration of 1 mg·mL^−1^. Three PLA_2_ (BdTX-I, BdTX-II and BdTX-III) found in *B. diporus*, also did not show hemolytic activity at the concentrations evaluated (10 µg·mL^−1^/ 0.8 μM). From the *B. jararacussu* venom, four PLA2 were obtained (SIIISPIIA, SIIISPIIB, SIIISPIIIA and SIIISPIIIB), and only SIIISPIIIB showed hemolytic activity at 13.3 µM (95.4%) [34].

Therefore, the high catalytic activity of PLA_2_ is not correlated with hemolytic activity, which is an important finding that supports its safe use as a model for bioprospection for new drugs. On the other hand, VRV-PL-VIIIa, from *Daboia russeli*, showed 100% hemolysis at 5 µg. A further study demonstrated that BmooPLA2, from *B. moojeni*, showed hemolytic activity at 0.07 µM. Indirect hemolytic activity was found in sheep blood, when incubated with 47.26 µM of a PLA_2_ from the venom of *B. alternatus*. This information confirms that the pharmacological activities present in PLA_2_ do not depend only on the catalytic activity [16,34,35,36,37,38].

The present study demonstrated that BmPLA2 has significant anticancer activity against Caco-2 (IC_50_ of 0.6 µM) and RD cells (no IC_50_), showing lower activity against the healthy strains MRC-5 and BV-2 (IC_50_ of 1.1 and 3.8 µM, respectively). The first investigation into the anticancer activity of snake venom dates from the 1970s, suggesting that Ancrod, a polypeptide from *Agkistrodon rhodostoma*, can generate defibrination, reducing the size of a tumor by fibrinolysis [39].

The svPLA_2_ strains MTX-I and MTX-II, from *B. brazili*, showed anticancer activity against Jurkat cells (leukemia cells), inhibiting 40% of the antitumor activity with 7.2 µM. BthA-I-PLA2, obtained from the venom of *B. jararacussu*, demonstrated antitumor activity against three different strains, showing 60% cytotoxicity against EAT (Erlich ascitic tumor), 50% against Jurkat (T-cell leukemia) and 30% against SKBr3 (human breast cancer), using a 7.4 µM (100 µg·mL^−1^) sample. BthTX-I, a PLA_2_ from *B. jararacussu*, presented an IC_50_ of 6.8 and 8 µM (81.2 and 104.35 µg·mL^−1^) against SK-BR-3 (human breast cancer cells) and MCF-7 (human breast cancer cells), respectively. In addition, in vivo experiments demonstrated that BthTX-I showed activity against S180 (sarcoma), reducing the tumor size by 76% after 60 days of treatment, when compared to the control [25,36,40,41].

From *Daboia russelii* venom, Drs-PLA2 was isolated and showed to be able to reduce the cell viability of human skin melanoma (SK-MEL-28) in a dose-dependent manner, with N IC_50_ of 0.02 µM. Another PLA_2_ with therapeutic potential against tumor cells, BthA-I- PLA2, from *B. jararacussu*, showed anticancer activity against human breast SK- BR-3 (20%), T-cell leukemia Jurkat (50%) and Erlich ascitic tumor EAT (70%), at 7.7 µM (100 µg·mL^−1^). From *B. mattogrossensis* venom, two PLA_2_, BmatTX-I and BmatTX- II, were obtained, showing activities against T-cell leukemia (Jurkat) (40 and 50% cytotoxic activity, respectively) and against human breast adenocarcinoma (SK-BR-3) (20% cytotoxic activity), at a concentration of 7.7 µM (100 µg·mL^−1^). From the *B. moojeni* toxin, BmooPLA2 showed cytotoxic activity against EAT and Jurkat strains, indicating 60 and 50% cellular growth reduction activity, respectively [10,36,42,43].

The svPLA_2_ have more than one possible mechanism suggesting the observed antitumor effect. The most observed mechanism, cytotoxicity, among others, occurs through the action of arachidonic acid (generated after the cleavage of phospholipids), which is cytotoxic both as a detergent and as an inducer of mitochondria permeability, as well as causing an uncoupling of oxidative phosphorylation. In addition, svPLA_2_ can affect cellular adhesion. Tumor growth can be facilitated by the expression of adhesion molecules (e.g., ICAM-1), considered a signal molecule. In contact with a svPLA_2_, a reduction in the expression and invasion of ICAM-1 in lung cancer cells was observed [44,45].

Another mechanism of PLA_2_ antitumor action is by acting on growth factors. For example, a svPLA2 from the venom of *Vipera ammodytes*, named as RVV-7, is able to bind to vascular endothelial growth factor (VEGF-A165), inhibiting the proliferative effect of VEGF on the endothelium. Treatment of melanoma cells (B16F10) with RVV-7 reduced tumor growth in guinea pigs [46].

PLA_2_ antitumor activities, as well as other antimicrobial activities, may be related to the interactions of the C-terminal region of the molecule with the cell membranes, usually described as capable of disrupting the hydrophilic matrix of membranes. Furthermore, PLA_2_ can act on the cell cycles of tumor lineages. BthTX-I was able to delay the G0/G1 phase of the cell cycle of PC-12 (pheochromocytoma) and B16F10 (melanoma) tumor cells, indicating that PLA_2_ interrupts mitotic progression, resulting in a reduction in cell duplication [25,40,41,47].

The SI results showed that BmPLA2 seems to be more selective in tumor cells, with values dependent on the non-tumorigenic lineages to be evaluated. When compared to MRC-5, BmPLA2 presents an SI of 1.35 for the Caco-2 strains. In addition, when compared to BV-2, the SI value is 6.33. The SI value of the NCI and RD lineage was not calculated as it did not have an IC_50_. This difference in SI values is observed because the healthy cells used in the experiment come from different tissues. The MRC-5 lineage comes from lung tissue composed of fibroblasts, while the BV-2 lineage comes from brain tissue, more specifically, microglial cells.

## 4. Materials and Methods

### 4.1. Bothrops Moojeni Venom, Cell Lines and Ethics Committee Guidelines

*B. moojeni* crude venom was collected from four adult specimens from the Serpentarium at the Universidade Católica Dom Bosco, Campo Grande, Mato Grosso do Sul, Brazil, by direct pressure on the snake venom glands. The cell lines used were as follows: the healthy human fibroblast cell line (MRC-5), the human colorectal cancer cell line (Caco-2), human rhabdomyosarcoma cells (RD) and mucoepidermoid carcinoma of the lung (NCI- H292), which were obtained from the Adolf Lutz Institute, and microglia cells (BV-2) were acquired from the Cell Bank of Rio de Janeiro. Mice (Mus musculus) erythrocytes were also used for assays, approved by the animal ethics committee from the Universidade Católica Dom Bosco under registration number 014/2018.

### 4.2. B. moojeni Venom Purification

Initially, *B. moojeni* crude venom (10 mg) was suspended in 1 mL of 50 mM ammonium bicarbonate buffer (Ambic), pH 7.8. Next, the sample was applied onto a molecular exclusion Sephacryl S-100 column (30 × 450 mm, Cytiva, Telangana, India) previously equilibrated with the same buffer, at room temperature. Fractions of 2.5 mL were collected at a flow rate of 0.625 mL·min^−1^. All fractions were monitored in a Biodrop (Biodrop Touch Duo Spectrophotometer) at 280 nm. The fractions corresponding to five different peaks were pooled and named F1 to F5.

Next, the peak 3 (F3) was applied onto a C18 column (Xterra MS 5 µm—4.6 × 250 mm) in HPLC (Waters e2695). The sample was dissolved with 0.1% TFA:water (*v*:*v*) and the elution was performed using a linear concentration gradient of 5–95% of solvent B (acetonitrile + 0.1% TFA) for 60 min at a flow rate of 1 mL·min^−1^. The protein profile was monitored at 216 nm. The four peaks obtained from F3 separated in Xterra MS 5 column were named F3.1 to F3.4.

As the final chromatographic step, the F3.4 peak was applied onto a C18 column (Xterra MS 5 µm—4.6 × 250 mm column) and separated in a stepwise gradient of solvent B (5% during 10 min, 5–40% 10–20 min, 40–60% 20–50 min and 60–95% 50–60 min). After this step, the sample purified was denominated BmPLA2.

### 4.3. SDS-PAGE Electrophoresis

Chromatographic fractions and purified BmPLA2 were analyzed by SDS-PAGE 12%, performed in Mini BioRad Gel Electrophoresis (Bio-Rad, Hercules, CA, USA), according to Laemmli (1970). The samples were incubated with a reducing sample buffer, containing β-mercaptoethanol. The gels were separated at 85 V and were stained with Coomassie brilliant blue R250. The PROMEGA^®^ Molecular weight marker (MW) was used (Laemmli, 1970).

### 4.4. Quantification of Proteins

The protein content from the *B. moojeni* crude venom, and in all fractions obtained through the chromatography steps, was determined using the Bradford method (Bradford, 1976). Bovine serum albumin (BSA) was used as the standard. The assays were carried out in triplicate.

### 4.5. Mass Spectrometry

BmPLA2 was analyzed using AutoFlex III matrix-assisted laser desorption ionization time-of-flight (MALDI-ToF) mass spectrometry equipment (Bruker Daltonics, Bremen, Germany) controlled by Flex Control 3.0 software (Bruker Daltonics, Bremen, Germany). The sample (15 μg·μL^−1^) was mixed with a sinapinic acid matrix solution (1:1, *v*:*v*), directly placed onto a target plate and dried at room temperature. The accurate mass of BmPLA2 was obtained in a positive linear mode, after external calibration, using the Protein Calibration Standard (Bruker Daltonics, Bremen, Germany). MALDI spectra were processed with Flex Analysis 3.0 software (Bruker Daltonics, Bremen, Germany).

### 4.6. Amino-Terminal Sequencing

The N-terminal primary sequence was determined using a Shimadzu PPSQ-31B/33B automated protein sequencer, performing Edman degradation. The equipment was previously calibrated with a phenylthiohydantoin (PTH) amino acids mixture standard and a sample from the lyophilized BmPLA2 was dissolved in 37% acetonitrile and applied on the PVDF membrane, dried under nitrogen flow and then submitted for analysis. The PTH amino acids were detected after separation on a RP-HPLC C18 column (4.6 × 250 mm), according to the manufacturer’s instructions. The resulting sequence was applied to NCBI protein BLAST search (BLASTP 2.8.0+) and the sequences that produced significant were aligned using CLUSTAL Omega (1.2.4).

### 4.7. Phospholipase Activity

The protocol used for the measurement of phospholipase activity was described by Holzer and Mackessy, with changes proposed by Serino-Silva and co-workers [48,49]. The BmPLA2 and fractions obtained during the purification were evaluated on a 96-well plate using 4-nitro-3-octanoyloxybenzoic acid (4N3OBA, Enzy Life Science, Oyster Bay, New York, NY, USA). Each well received 200 µL of 10 mM Tris-HCl buffer, containing 10 mM CaCl2 and 100 mM NaCl, pH 7.8. Then, 20 µL of 4N3OBA substrate (1 mg·mL^−1^, dissolved in acetonitrile), 20 µL of water and 20 µL of the source of enzyme (crude venom, F3 and BmPLA2) were added. As a positive control, a commercial PLA_2_ (Phospholipase A_2_ from bovine pancreas—P9913 Sigma) was used, and the negative control was performed with BSA, a non-enzymatic protein. The reaction was developed at 37 °C, with readings at 10 min intervals, with a total of 150 min, determined at 425 nm. The assay was carried out in triplicate.

### 4.8. Hemolytic Assay

Mouse erythrocyte blood was collected and stored at 4 °C until use. The cells were washed three times with a 50 μM phosphate buffer, pH 7.4. The erythrocyte suspension was incubated with *B. moojeni* venom fractions in serial dilution (512 to 16 μg·mL^−1^). From the molecular mass of BmPLA2, these concentrations are equivalent to 37 to 1.2 μM in 100 mL final volume. The samples were incubated at room temperature for 60 min. After centrifugation at 3000 rpm, hemoglobin release was monitored by reading the supernatant at 425 nm in a Spectramax microplate reader. The negative control of hemolysis was performed with erythrocytes suspended in a 50 mM phosphate buffer, pH 7.4. The positive control for hemolysis was carried out with 1% triton X-100 solution dissolved in distilled water. The assays were performed in triplicate.

### 4.9. Anticancer Activity

#### 4.9.1. Cell Culture

Cell lines were cultivated at the Immunology Laboratory (UCDB). MRC-5 and RD cells were maintained in DMEM high glucose medium, BV-2 and NCI-H292 cells were maintained in RPMI-1640 medium and the Caco-2 lineage was maintained in 199 medium. All media were supplemented with 10% fetal bovine serum (FBS) and for the MRC-5, RD, BV-2, NCI strains were added 100 µg·mL^−1^ penicillin and 100 µg·mL^−1^ streptomycin (Gibco, São Paulo, Brazil) at 37 °C in an oven with 5% CO_2_.

#### 4.9.2. Cytotoxicity by MTT Assay

To verify the anticancer activity, the viability of cancer cells for NCIH292, RD and Caco-2, and normal cells MRC-5 and BV-2, were evaluated according to the method adapted from Mosmann (1983), based on the enzymatic reduction of the 3- (4,5-demethylthiazol-2-yl)-2,5-diphenyltetrazolium bromide (MTT, Sigma, São Paulo, Brazil) to formazan crystals. Cells were plated at 1 × 104 cells/well-1 in 96-well microplates and treated with 100 µL of different concentrations of BmPLA2 (9.25, 4.62, 2.31, 1.15, 0.57, 0.28 and 0.14 µM) for 24 h. As a negative control, culture medium was used. After the incubation period, the supernatant was removed and 100 µL of MTT solution (1 mg·mL^−1^ diluted in culture medium) was added to the cells. After 4 h of incubation, the formazan crystals were resuspended with 100 µL of dimethyl sulfoxide (DMSO) and read at 570 nm in a Thermo scientific reader (Model MultiSkan Go). Three independent experiments were carried out in triplicate. Cell viability was calculated from the following formula:Cell viability (%) = (Abs_sample_ ÷ Abs_negative control_) × 100

#### 4.9.3. Statistics

All assays were carried out in triplicate and the results were expressed as the mean ± S.E.M. Differences between treatments and controls were analyzed by One-Way ANOVA followed by the Dunnet and Bonferroni post-test, using the software GraphPad Prism (GraphPad Software, Inc., San Diego, CA, USA).

## 5. Conclusions

PLA_2_ isolated from snake venoms is a multifunctional set of proteins displaying great potential for biotechnological application in the medical field. Purification of BmPLA2 from *Bothrops moojeni* venom resulted in a new protein of 13,868 Da, sharing a high sequential identity with other types of PLA_2_. In addition to the enzymatic activity, very low hemolytic activity was observed against murine erythrocytes, which makes this novel PLA_2_ safe to use as a potential pharmacological compound for cancer treatment. Finally, anticancer activity against Caco-2 and RD cells was observed, presenting an SI that indicates the promising application of BmPLA2 for this purpose, supporting further studies and future structural modifications of the molecule.

## Figures and Tables

**Figure 1 pharmaceuticals-15-00724-f001:**
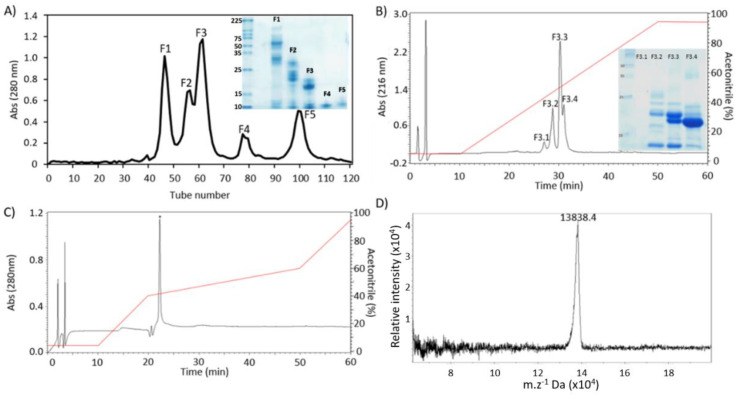
Purification steps of BmPLA2 from *B. moojeni* venom. (**A**) Separation profile of *B. moojeni* venom on Sephacryl S-100 column using 50 mM Ambic, pH 7.8. Inset: 12% SDS-PAGE showing peaks F1 to F5. (**B**) Profile of F3 peak on a RP-HPLC C18 column. Elution was performed using a linear gradient 5–95% of solvent B, for 60 min, at a flow rate of 1 mL·min^−1^. Inset, 12% SDS-PAGE of peaks F3.1 to F3.4. (**C**) Profile of F3.4 on a RP-HPLC C18 column: Elution was performed using a stepwise gradient: 5% of solvent B to 10 min, 5–40% to 10–20 min, 40–60% to 20–50 min, 60–95% to 50–60 min at a flow rate of 1 mL·min^−1^. (**D**) Molecular mass of BmPLA2, 13,838.4 Da, determined by MALDI-ToF.

**Figure 2 pharmaceuticals-15-00724-f002:**
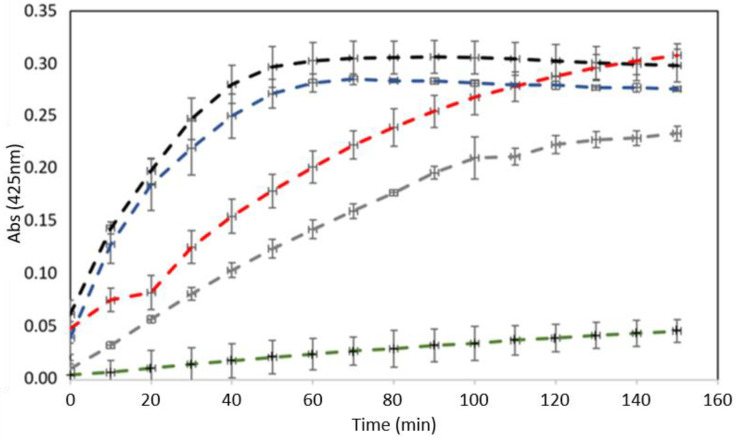
Enzymatic activity of BmPLA2 incubated with specific substrate (4N3OBA). Legend color: BSA, negative control (green); commercial PLA_2_ (gray); *B. moojeni* crude venom (black); peak F3 from Sephacryl S-100 (blue); and BmPLA2 (red).

**Figure 3 pharmaceuticals-15-00724-f003:**
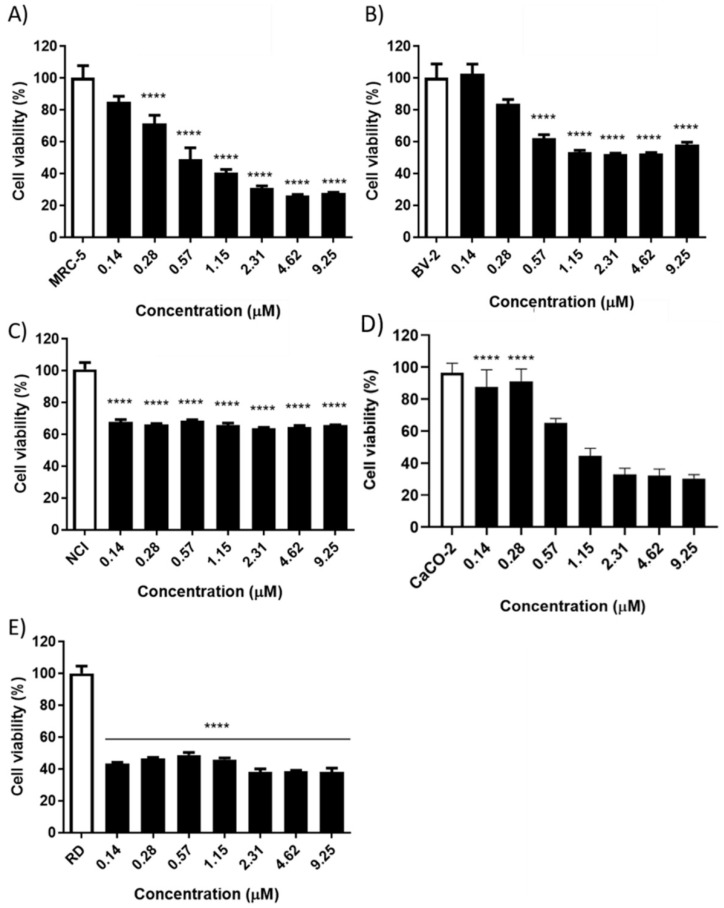
Assay of cellular viability investigating the effects of BmPLA2 on cancer and healthy cells. (**A**) Healthy human fibroblast cell line (MRC-5); (**B**) Microglia cell line; (**C**) Mucoepidermoid carcinoma of the lung (NCI-H292); (**D**) Human colorectal cancer cell line (Caco-2); (**E**) Human rhabdomyosarcoma cells (RD). All cell lines were treated with BmPLA2 (9.25, 4.62, 2.31, 1.15, 0.57 and 0.28 µM) for 24 h. The viability was determined using MTT reagent. All data were expressed as mean ± S.E.M and procedures were carried out in duplicate. **** means differences between the groups were considered statistically significant.

**Table 1 pharmaceuticals-15-00724-t001:** Multiple alignment of PLAs N-terminal demonstrated region of similarity with BmPLA. PLA_2_ from *B. moojeni* venom; Tgc-E6 from *Tantilla gracilis* venom and D1E6b from *Cerrophidion godmani* venom, with a 77% identity. Positions with fully conserved residue (*); One of the high-scoring groups is conserved (:).

Species	PLA_2_	Sequence	Identity (%)
*B. moojeni*	BmPLA	FKWQFEMLIMKIAKTSGFMFYSSY	-
*B. moojeni*	BMOOPLA2	NLWQFEMLIMKIAKTSGFLFYSSY	95
*T. gracilis*	TGC-E6	SLMQFEMLIMKLAKSSGMFWYSAY	77
*C. godmani*	D1E6b	DLIQFEMLIMKVAKRSGMFWYSAY	77
		********:** **:::**:*	

**Table 2 pharmaceuticals-15-00724-t002:** IC_50_ and selectivity index of MRC-5, BV-2, Caco-2 and RD for BmPLA2.

Cell Line	IC_50_ (µM)	IS * (MRC-5)	IS * (BV-2)
MRC-5	1.1	-	-
BV-2	3.8	-	-
Caco-2	0.6	1.85	6.33

* Selectivity index = ratio between IC_50_ healthy cell line (µM) and cancer cell line IC_50_ (µM).

## Data Availability

Data is contained within the article.

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
