# Peer review of "Purification, Characterization and Evaluation of the Antitumoral Activity of a Phospholipase A2 from the Snake Bothrops moojeni"

_pharmaceuticals, 2022, doi:10.3390/ph15060724_

Round 1

Reviewer 1 Report

Dear Authors,

I reviewed your manuscript entitled: Purification characterization and evaluation of the antitumoral activity of a Phospholipase A2 from the snake Bothrops moojeni.  I think it is well designed and well written and no correction is needed. 

The point that I should mention just for your realization, I suggest you identify the full sequences of BmPLA2 that will be complimentary work.

Anyhow, I endorse it for publication in  Pharmaceuticals.

Best Regards

Author Response

Reviewer #1

1 - I reviewed your manuscript entitled: Purification characterization and evaluation of the antitumoral activity of a Phospholipase A2 from the snake Bothrops moojeni. I think it is well designed and well written and no correction is needed.

Response: The authors thank you for your time in correcting the paper.

2 - The point that I should mention just for your realization, I suggest you identify the full sequences of BmPLA2 that will be complimentary work.

Response: The authors agree with the reviewer, however, to perform the partial sequencing of the molecule, Edman degradation was used. Due to the large size of the molecule, the methodology used is not able to generate the full sequence phospholipase. Therefore, the idea was to characterize one more biochemical parameter to confirm the enzyme. In addition, the authors intend to continue the work, and perform the complete sequencing, as well as the resolution of the three-dimensional structure for conformational studies.

Reviewer 2 Report

The submitted manuscript describes the isolation and partial characterisation of a novel PLA2 from the well known and studied Bothrops moojeni venom. The new protein has different selectivity over other known PLA2's from the same venom and has reasonable selectivity for cancer cells lines. This result would be of interest to those researchers developing novel anticancer agents or those requiring specific pharmacological tools. I recommend publication but there are a number of things that could be improved in the article to make it more suitable for the audience of the journal and also improve its overall quality.

  1. The introduction is lacking in a thorough background on the large amount of PLA2s previously found in this venom and in others, specifically there is well know differences between the acidic and basic/neutral PLAs that should be mentioned here and the known anticancer properties of the various subtypes. The paragraph in lines 73-76 must be expanded to further introduce these key previous findings as without them the reader has little context of previous knowledge.
  2. Similarly it would be much better for the known anticancer actions of these PLAs to be better described in the introduction and/or discussion and I found it lacking in explanation on the exact nature of the PLAs mode of action.
  3. The gel diagram in F1A should be improved as it is of low resolution
  4. The overall mass and partial sequence has been determined for the protein. Of course it would have been considerably better to have the full protein sequence as without it drug design may be limited. Can the authors expand on the proteins similarity to other PLAs - specifically BMOOPLA2, which appears most similar. What is known of the catalytic regions or binding sites of the two proteins - has sufficient sequence been reported to give others useful information to derive smaller peptide-like molecules which may have activity.
  5. I find the last sentence of section 3 confusing. The activity is lower in the microglia cell line so therefore the compounds could be better used there? One would need to know far more about structural aspects of the protein to limit toxicity in the other cell lines before considering the least active cell line as a target - if this is not the argument of the authors then the sentence needs rewriting to make it clearer.
  6. How did the authors calculate the IC50 for RD cell line with the data shown, all values are below 50 and statistically indifferent?

Author Response

Reviewer #2

1 - The submitted manuscript describes the isolation and partial characterisation of a novel PLA2 from the well known and studied Bothrops moojeni venom. The new protein has different selectivity over other known PLA2's from the same venom and has reasonable selectivity for cancer cells lines. This result would be of interest to those researchers developing novel anticancer agents or those requiring specific pharmacological tools. I recommend publication but there are a number of things that could be improved in the article to make it more suitable for the audience of the journal and also improve its overall quality.

Response: The authors thank you for your time in correcting the paper.

2 - The introduction is lacking in a thorough background on the large amount of PLA2s previously found in this venom and in others, specifically there is well know differences between the acidic and basic/neutral PLAs that should be mentioned here and the known anticancer properties of the various subtypes.

Response: The authors agree with the reviewer and to clarify the idea more examples of phospholipases with antitumor activity were added in the introduction. However, the differences between acid/base PLAs were not explored, as this was not the focus of the work, and even in the literature, we have examples of acidic PLAs with cytotoxic activity (BmooTX-I and MTX-I) and acid PLAs without activity cytotoxic (BJ-PLA2-I and BEI-PLA2), indicating that the antitumor activity of PLAs does not necessarily depend on this physicochemical parameter.

Now reads:

(Introduction, page 3, line 80-88):

Previous reports demonstrated that Bothrops svPLA2 from snake venoms present anti-tumor activity. BthA-I-PLA2 from B. jararacussu showed antitumor activity in leukemia (Jurkat), human breast tumor (BR-3) and Ehrlich ascites (EAT) lines, BmooTX-I (B. moojeni) and MTX-I (B. brazili) indicating activity against leukemic lineage (Jurkat); Myotoxin III (B. asper) demonstrating cytotoxicity in adrenal tumor. In addition, other svPLA2 presented anti-tumor activity, like PLA2 RVV-7 (Daboia russeli) showing activity against melanoma (B16F10), MVL-PLA2 (Macrovipera lebetina), with activity against fibrosarcoma, melanoma, adenocarcinoma and leukemia, and F1 CTX (Crotalus durissus terrificus) indicating cytotoxicity against cervical and esophagus cancer cell lines [9-13].”

References:

  1. Azevedo, F. V. P. V.; Lopes, D. S.; Cirilo Gimenes, S. N.; Achê, D. C.; Vecchi, L.; Alves, P. T.; Guimarães, D. de O.; Rodrigues, R. S.; Goulart, L. R.; Rodrigues, V. de M.; Yoneyama, K. A. G. Human Breast Cancer Cell Death Induced by BnSP-6, a Lys-49 PLA2 Homologue from Bothrops Pauloensis Venom. Int. J. Biol. Macromol. 2016, 82, 671–677.
  2. Cedro, R. C. A.; Menaldo, D. L.; Costa, T. R.; Zoccal, K. F.; Sartim, M. A.; Santos-Filho, N. A.; Faccioli, L. H.; Sampaio, S. V. Cytotoxic and Inflammatory Potential of a Phospholipase A2 from Bothrops Jararaca Snake Venom. J. Venom. Anim. Toxins Incl. Trop. Dis. 2018, 24 (1), 1–14.
  3. Daniele, J. J.; Bianco, I. D.; Delgado, C.; Carrillo, D. B.; Fidelio, G. D. A New Phospholipase A2 Isoform Isolated from Bothrops Neuwiedii (Yarara Chica) Venom with Novel Kinetic and Chromatographic Properties. Toxicon 1997, 35 (8), 1205–1215.
  4. de Vasconcelos Azevedo, F. V. P.; Zóia, M. A. P.; Lopes, D. S.; Gimenes, S. N.; Vecchi, L.; Alves, P. T.; Rodrigues, R. S.; Silva, A. C. A.; Yoneyama, K. A. G.; Goulart, L. R.; de Melo Rodrigues, V. Antitumor and Antimetastatic Effects of PLA2-BthTX-II from Bothrops Jararacussu Venom on Human Breast Cancer Cells. Int. J. Biol. Macromol. 2019, 135, 261–273.
  5. Stábeli, R. G.; Amui, S. F.; Sant’Ana, C. D.; Pires, M. G.; Nomizo, A.; Monteiro, M. C.; Romão, P. R. T.; Guerra-Sá, R.; Vieira, C. A.; Giglio, J. R.; Fontes, M. R. M.; Soares, A. M. Bothrops Moojeni Myotoxin-II, a Lys49-Phospholipase A2 Homologue: An Example of Function Versatility of Snake Venom Proteins. Comp. Biochem. Physiol. - C Toxicol. Pharmacol. 2006, 142 (3-4 SPEC. ISS.), 371–381.

3 - The paragraph in lines 73-76 must be expanded to further introduce these key previous findings as without them the reader has little context of previous knowledge.

Response: The authors agree with reviewer suggestion and more additional information, such as the mechanism of action and the description of some more phospholipases with antitumor activity were added.

Now reads:

(Introduction, page 3-4, line 71-88):

Due to their role in a large number of human inflammatory diseases, PLA2 shows a medical-scientific interest. The classification of this protein group occurs according to the site of hydrolysis where, PLA2 are enzymes that catalyze the hydrolysis of phospholipids at the sn-2 position, releasing free fatty acids, arachidonic acid and lysophospholipids. By consequence, they play important roles in the metabolism of dietary and structural lipids in cell membranes. Hydrolysis of the lipids in cell membranes leads to loss of their structure, impairing their selective permeability. As a consequence of this activity, svPLA2 has a diverse anti-microbial activity, being able to act on cultures of pathogenic bacteria, fungi, protozoa, viruses and tumor lineages [8].

Previous reports demonstrated that Bothrops svPLA2 from snake venoms present anti-tumor activity. BthA-I-PLA2 from B. jararacussu showed antitumor activity in leukemia (Jurkat), human breast tumor (BR-3) and Ehrlich ascites (EAT) lines, BmooTX-I (B. moojeni) and MTX-I (B. brazili) indicating activity against leukemic lineage (Jurkat); Myotoxin III (B. asper) demonstrating cytotoxicity in adrenal tumor. In addition, other svPLA2 presented anti-tumor activity, like PLA2 RVV-7 (Daboia russeli) showing activity against melanoma (B16F10), MVL-PLA2 (Macrovipera lebetina), with activity against fibrosarcoma, melanoma, adenocarcinoma and leukemia, and F1 CTX (Crotalus durissus terrificus) indicating cytotoxicity against cervical and esophagus cancer cell lines [9-13].”

References

  1. Quach, N. D.; Arnold, R. D.; Cummings, B. S. Secretory Phospholipase A2 Enzymes as Pharmacological Targets for Treatment of Disease. Biochem. Pharmacol. 2014, 90 (4), 338–348.
  2. Azevedo, F. V. P. V.; Lopes, D. S.; Cirilo Gimenes, S. N.; Achê, D. C.; Vecchi, L.; Alves, P. T.; Guimarães, D. de O.; Rodrigues, R. S.; Goulart, L. R.; Rodrigues, V. de M.; Yoneyama, K. A. G. Human Breast Cancer Cell Death Induced by BnSP-6, a Lys-49 PLA2 Homologue from Bothrops Pauloensis Venom. Int. J. Biol. Macromol. 2016, 82, 671–677.
  3. Cedro, R. C. A.; Menaldo, D. L.; Costa, T. R.; Zoccal, K. F.; Sartim, M. A.; Santos-Filho, N. A.; Faccioli, L. H.; Sampaio, S. V. Cytotoxic and Inflammatory Potential of a Phospholipase A2 from Bothrops Jararaca Snake Venom. J. Venom. Anim. Toxins Incl. Trop. Dis. 2018, 24 (1), 1–14.
  4. Daniele, J. J.; Bianco, I. D.; Delgado, C.; Carrillo, D. B.; Fidelio, G. D. A New Phospholipase A2 Isoform Isolated from Bothrops Neuwiedii (Yarara Chica) Venom with Novel Kinetic and Chromatographic Properties. Toxicon 1997, 35 (8), 1205–1215.
  5. de Vasconcelos Azevedo, F. V. P.; Zóia, M. A. P.; Lopes, D. S.; Gimenes, S. N.; Vecchi, L.; Alves, P. T.; Rodrigues, R. S.; Silva, A. C. A.; Yoneyama, K. A. G.; Goulart, L. R.; de Melo Rodrigues, V. Antitumor and Antimetastatic Effects of PLA2-BthTX-II from Bothrops Jararacussu Venom on Human Breast Cancer Cells. Int. J. Biol. Macromol. 2019, 135, 261–273.
  6. Stábeli, R. G.; Amui, S. F.; Sant’Ana, C. D.; Pires, M. G.; Nomizo, A.; Monteiro, M. C.; Romão, P. R. T.; Guerra-Sá, R.; Vieira, C. A.; Giglio, J. R.; Fontes, M. R. M.; Soares, A. M. Bothrops Moojeni Myotoxin-II, a Lys49-Phospholipase A2 Homologue: An Example of Function Versatility of Snake Venom Proteins. Comp. Biochem. Physiol. - C Toxicol. Pharmacol. 2006, 142 (3-4 SPEC. ISS.), 371–381.

4 - Similarly it would be much better for the known anticancer actions of these PLAs to be better described in the introduction and/or discussion and I found it lacking in explanation on the exact nature of the PLAs mode of action.

Response: The authors followed this suggestion, and the general mechanism of actions for phospholipases were added in the Introduction, and in the Discussion topic.

Now reads:

(Introduction, page 2-3, line 71-79)

Due to their role in a large number of human inflammatory diseases, PLA2 shows a medical-scientific interest. The classification of this protein group occurs according to the site of hydrolysis where, PLA2 are enzymes that catalyze the hydrolysis of phospholipids at the sn-2 position, releasing free fatty acids, arachidonic acid and lysophospholipids. By consequence, they play important roles in the metabolism of dietary and structural lipids in cell membranes. Hydrolysis of the lipids in cell membranes leads to loss of their structure, impairing their selective permeability. As a consequence of this activity, svPLA2 has a diverse anti-microbial activity, being able to act on cultures of pathogenic bacteria, fungi, protozoa, viruses and tumor lineages [8].

(Discussion, page 8, lines 266-278)

The svPLA2 have more than one possible mechanism suggesting the observed antitumor effect. The most observed mechanism, cytotoxicity, among other ways, occurs through the action of arachidonic acid (generated after cleavage of phospholipids), which is cytotoxic both as a detergent and as an inducer of mitochondria permeability, as well as causing an uncoupling of oxidative phosphorylation. In addition, svPLA can affect cellular adhesion. A tumor growth can be facilitated by the expression of adhesion molecules (e.g. ICAM-1), considered a signal molecule. In contact with a svPLA2, a reduction of the expression and invasion of ICAM-1 in lung cancer cells was observed. Another mechanism of PLA2 antitumor action is by acting on growth factors. For example, a svPLA2 from the venom of Vipera ammodytes, named as RVV-7, is able to bind to vascular endothelial growth factor (VEGF-A165), inhibiting the proliferative effect of VEGF on endothelium. Treatment of melanoma cells (B16F10) with RVV-7 reduced tumor growth in guinea pigs [46,47].”.

References:

  1. Quach, N. D.; Arnold, R. D.; Cummings, B. S. Secretory Phospholipase A2 Enzymes as Pharmacological Targets for Treatment of Disease. Biochem. Pharmacol. 2014, 90 (4), 338–348.
  2. Cummings, B. S. Phospholipase A2 as Targets for Anti-Cancer Drugs. Biochem. Pharmacol. 2007, 74 (7), 949–959.
  3. Yu, J. A.; Kalatardi, S.; Dohse, J.; Sadaria, M. R.; Meng, X.; Fullerton, D. A.; Weyant, M. J. Group IIa SPLA2 Inhibition Attenuates NF-ΚB Activity and Promotes Apoptosis of Lung Cancer Cells. Anticancer Res. 2012, 32 (9), 3601–3607.

5 - The gel diagram in F1A should be improved as it is of low resolution

Response: The authors agree with the reviewer and the resolution of the electrophoresis gel image was improved.

6 - The overall mass and partial sequence has been determined for the protein. Of course it would have been considerably better to have the full protein sequence as without it drug design may be limited.

Response: The suggestion is excellent, however the proposal will be realized in the future. Edman degradation methodology used is not able to generate the complete phospholipase sequence due to the large size of the molecule. Therefore, the aim was to characterize one more biochemical parameter to confirm the enzyme.

7 - Can the authors expand on the proteins similarity to other PLAs - specifically BMOOPLA2, which appears most similar.

Response: In addition to citing the molecular mass similarity between BmPLA2 and the other PLA2s (Discussion, lines 196-204), the authors described the similarities in the percentage of acetonitrile required to elute a svPL2 from RP-HPLC methodology (Discussion, lines 180-195). As additional information, from the reviewer's suggestion, the antitumor activity of BmooPLA2 was added, the svPLA2 described in the sequencing table (Table 1).

Now reads:

(Discussion, page 7-8, lines 255-265)

From Daboia russelii venom, Drs-PLA2 was isolated, and showed to be able to reduce the cell viability of human skin melanoma (SK-MEL-28), in a dose-dependent manner, with IC50 of 0.02 µM. Another PLA2 with therapeutic potential against tumor cells, BthA-I- PLA2, from B. jararacussu, showed anticancer activity against human breast SK- BR-3 (20%), T-cell leukemia Jurkat (50%) and Erlich ascitic tumor EAT (70%), at 7.7 µM. From B. mattogrossensis venom, two PLA2, BmatTX-I and BmatTX- II, were obtained, showing activities against T-cell leukemia (Jurkat) (40 and 50% cytotoxic activity, respectively) and against human breast adenocarcinoma (SK-BR-3) (20% cytotoxic activity), at a concentration of 7.7 µM. From B. moojeni toxin, BmooPLA2 showed cytotoxic activity against EAT and Jurkat strains, indicating 60 and 50% cellular growth reduction activity, respectively [10,38,44,45].”

References

  1. Cedro, R. C. A.; Menaldo, D. L.; Costa, T. R.; Zoccal, K. F.; Sartim, M. A.; Santos-Filho, N. A.; Faccioli, L. H.; Sampaio, S. V. Cytotoxic and Inflammatory Potential of a Phospholipase A2 from Bothrops Jararaca Snake Venom. J. Venom. Anim. Toxins Incl. Trop. Dis. 2018, 24 (1), 1–14.
  2. Silveira, L. B.; Marchi-Salvador, D. P.; Santos-Filho, N. A.; Silva Jr, F. P.; Marcussi, S.; Fuly, A. L.; Nomizo, A.; da Silva, S. L.; Stábeli, R. G.; Arantes, E. C. Isolation and Expression of a Hypotensive and Anti-Platelet Acidic Phospholipase A2 from Bothrops Moojeni Snake Venom. J. Pharm. Biomed. Anal. 2013, 73, 35–43.
  3. Moura, A. A. D.; Kayano, A. M.; Oliveira, G. A.; Setúbal, S. S.; Ribeiro, J. G.; Barros, N. B.; Nicolete, R.; Moura, L. A.; Fuly, A. L.; Nomizo, A.; Da Silva, S. L.; Fernandes, C. F. C.; Zuliani, J. P.; Stábeli, R. G.; Soares, A. M.; Calderon, L. A. Purification and Biochemical Characterization of Three Myotoxins from Bothrops Mattogrossensis Snake Venom with Toxicity against Leishmania and Tumor Cells. Biomed Res. Int. 2014, 2014.
  4. Khunsap, S.; Pakmanee, N.; Khow, O.; Chanhome, L.; Sitprija, V.; Suntravat, M.; Lucena, S. E.; Perez, J. C.; Sánchez, E. E. Purification of a Phospholipase A2 from Daboia Russelii Siamensis Venom with Anticancer Effects. J. Venom Res. 2011, 2, 42.

8 - What is known of the catalytic regions or binding sites of the two proteins - has sufficient sequence been reported to give others useful information to derive smaller peptide-like molecules which may have activity.

Response: The N-terminal amino acid portion of BmPLA described in this article presents information that confirm the phopholipase enzyme together molecular mass and mainly the catalytic activity on specific substrate. For the next steps, we pretend to sequence fully BmPLA, with the intention of promote structural studies from BmPLA.

9 - I find the last sentence of section 3 confusing.

Response: The authors agree with the reviewer and to better understanding, this sentence was removed.

10 - The activity is lower in the microglia cell line so therefore the compounds could be better used there?

Response: The authors agree with reviewer and about the cell lines used as controls, BmPLA showed lower activity against mycroglia cells (BV-2) than against lung cells (MRC-5). This could indicate a possible application of BmPLA in tissues similar to BV-2. However, to confirm this suggestion, further studies would be needed, and because of this, this sentence was removed.

11 - One would need to know far more about structural aspects of the protein to limit toxicity in the other cell lines before considering the least active cell line as a target - if this is not the argument of the authors then the sentence needs rewriting to make it clearer.

Response: The authors agree with the comment and, as already described in the previous two comments, this statement was removed.

12 - How did the authors calculate the IC50 for RD cell line with the data shown, all values are below 50 and statistically indifferent?

Response: The authors thank you for the comment, because the calculation of IC50 for the RD strain was performed erroneously. Because of this, this information has been updated and the text was corrected.

Now reads:

(Results, Page 6, Lines 166-172)

“Based on the IC50 obtained for cancer and healthy cells, we calculated the selectivity index (SI) (Table 2). Using MRC-5 as a comparison, in the Caco-2 strains, the SI values were 1.85, while when comparing with the BV2, the SI values were 6.33, suggesting that BmPLA2 possesses selectivity for cancer cell membranes in comparison with healthy membranes. The SI values for the NCI and RD strains were not calculated because the IC50 of these strains could not be obtained. To obtain this value in the RD strain, it would be necessary to repeat the evaluation, using lower concentrations of the BmPLA.”

(Discussion, Page 8, Lines 286-290)

“The SI results showed that BmPLA2 seems to be more selective in tumor cells, with values dependent on the non-tumorgenic lineages to be evaluated. When compared to MRC-5, BmPLA2 presents an SI of 1.35 for the Caco-2 strains. In addition, when compared to BV-2, the SI value are 6.33. The SI value of the NCI and RD lineage was not calculated as it did not have an IC50.”.